# Selective Recognition of Gallic Acid Using Hollow Magnetic Molecularly Imprinted Polymers with Double Imprinting Surfaces

**DOI:** 10.3390/polym14010175

**Published:** 2022-01-02

**Authors:** Jiawei Li, Xinji Zhou, Yu Yan, Dianling Shen, Danqing Lu, Yaping Guo, Lianwu Xie, Bin Deng

**Affiliations:** 1College of Sciences, Central South University of Forestry and Technology, Changsha 410004, China; lijiawei9568@163.com (J.L.); z1778789514@126.com (X.Z.); yy433130@126.com (Y.Y.); a18711500236@126.com (D.S.); danqinglu@csuft.edu.cn (D.L.); guoyaping@csuft.edu.cn (Y.G.); 2College of Chemistry Biology and Environmental Engineering, Xiangnan University, Chenzhou 423043, China

**Keywords:** hollow magnetic molecularly imprinted polymers, gallic acid, double imprinting surfaces, selective recognition, adsorption

## Abstract

Gallic acid is widely used in the field of food and medicine due to its diversified bioactivities. The extraction method with higher specificity and efficiency is the key to separate and purify gallic acid from complex biological matrix. Herein, using self-made core-shell magnetic molecularly imprinted polymers (MMIP) with gallic acid as template, a hollow magnetic molecularly imprinted polymer (HMMIP) with double imprinting/adsorption surfaces was prepared by etching the mesoporous silica intermediate layer of MMIP. The characterization and adsorption research showed that the HMMIP had larger specific surface area, higher magnetic response strength and a more stable structure, and the selectivity and saturated adsorption capacity (2.815 mmol/g at 318 K) of gallic acid on HMMIP were better than those of MMIP. Thus, in addition to MMIP, the improved HMMIP had excellent separation and purification ability to selectively extract gallic acid from complex matrix with higher specificity and efficiency.

## 1. Introduction

Gallic acid (GA) is a polyphenolic hydroxybenzoic acid that is derived from plants such as *Cornus officinallis*. In recent years, GA has attracted much attention and is widely used in the field of food and medicine due to its diversified bioactivities [1,2,3]. It has many medical applications, such as regulating the proliferation and migration of vascular smooth muscle cells to prevent atherosclerosis [4], and inhibiting the proliferation of prostate cancer cells [5]. It can also be used as an effective drug ingredient against dengue virus type 2 [6].

The traditional methods for extracting GA mainly include solvent extraction [7], solid-phase extraction [8], ultrasonic-assisted extraction [9], supercritical fluid extraction [10,11], and many other techniques. The traditional methods for extracting GA are time-consuming and labor-intensive. The extraction method with the highest specificity and efficiency is the key to separating and purifying GA from complex biological matrix. We noticed that molecularly imprinted polymer (MIP) is one type of effective adsorbent to separate the target from complex samples.

The mechanism of MIP synthesis (non-covalent strategy) is shown in Appendix A. Through the action of covalent bonds or non-covalent bonds, the target and the functional monomer form a spatial complementary structure, and then the crosslinking agent and initiator are added to generate polymerization to obtain a high crosslinking polymer with certain mechanical strength. Finally, the target is eluted to obtain the MIP with recognition sites.

The origin of molecular imprinting technology has a long history, dating back at least to the 1970s [12]. The establishment of the Society for Molecular Imprinting made molecular imprinting technology grow rapidly. At present, molecular imprinting technology has shown great brilliance in many aspects, such as biomedical analyses [13], electrochemical sensor preparation [14,15], and extraction of active components from natural products [16]. For the extraction of active components from natural products, most of them use core-shell molecularly imprinted polymers as adsorbents. Compared with conventional extraction methods, it has the advantages of higher stability, higher selectivity [17], and higher adsorption capacity [18]. If the magnetic material is introduced as the core, MIP can be separated from the sample solution quickly once using an external magnet. However, there is only one layer of adsorption surface with limited specific surface area in the core-shell molecularly imprinted polymer. If the inner surface of the MIP layer could be utilized as an adsorption surface, double adsorption surfaces including the inner and outer MIP layer should be realized to increase the specific surface area to achieve higher adsorption capacity. There have been a a few studies on the exploration of hollow structures. For example, Kang et al. [19] prepared a hollow Fe_3_O_4_ nanosphere for silybin adsorption. Wang et al. [20] prepared magnetic hollow molecularly imprinted polymers (M-H-MIPs) for the detection of triazines in food samples by in-situ growth of magnetic Fe_3_O_4_ nanoparticles on the surface of M-H-MIPs through multi-step swelling polymerization.

Therefore, the surface molecular imprinting method was adopted to prepare a novel adsorbent for the separation and purification of GA in this paper. With the magnetic Fe_3_O_4_ nanoparticle as the core and a layer of mesoporous SiO_2_ (mSiO_2_) as the intermediate layer, the molecularly imprinted layer was polymerized on the outer surface to obtain the core-shell magnetic molecularly imprinted polymer Fe_3_O_4_@mSiO_2_@MIP (MMIP). Then, mSiO_2_ was etched with HF acid to form the hollow molecularly imprinted polymer Fe_3_O_4_@Hollow MIP (HMMIP). HMMIP has double adsorption surfaces on the inner and outer MIP layers, which is different than the traditional core-shell molecularly imprinted polymers (they usually have only the outermost adsorption surface). The structure of HMMIP was characterized and the adsorption performance of HMMIP for GA was investigated through kinetic, thermodynamic, and specific adsorption assessments. All the resultant data are better than traditional core-shell MMIP.

## 2. Experimental

### 2.1. Chemicals and Apparatus

Methacryloxy propyl trimethoxyl silane (MPS, 97.0%), 4-vinylpyridine (4-VP, 96.0%), ethylene glycol dimethacrylate (EGDMA, 98.0%), and 2,2′-azobis-isobutyronitrile (AIBN, 98.0%) were supplied by Shanghai McLean Biochemical Technology Co., Ltd. (Shanghai, China). Absolute ethyl alcohol (≥99.7%), acetonitrile (≥99.8%), ethylene glycol (99.0%), methanol (99.8%), cetyltrimethyl ammonium bromide (CTAB, 99.0%), tetraethyl orthosilicate (TEOS, 99.0%), ferric trichloride hexahydrate (FeCl_3_·_6_H_2_O, 99.0%), anhydrous sodium acetate (99.0%), glacial acetic acid (99.8%), acetone (99.5%), polyethylene glycol 6000 (99.0%), and hydrofluoric acid (HF, ≥40.0%) were supplied by Sinopharm Chemical Reagents Co., Ltd. (Beijing, China). Gallic acid (GA, 99.0%) was supplied by J&K Scientific Ltd. (Beijing, China). The solutions used were prepared with deionized water and none of the chemicals underwent any further purification.

Transmission electron microscopy (TEM), particle size analysis, Fourier transform–infrared spectroscopy (FT-IR), X-ray photoelectron spectroscopy (XPS), a vibrating sample magnetometer (VSM), the Brunauer–Emmett–Teller specific surface area test method (BET), and thermogravimetry analysis (TGA) were used to characterize the structure of HMMIP and MMIP. TEM (JEOL JEM 2100, Tokyo, Japan) was used to observe the coating structure, diameter, and appearance. FT-IR (Thermo Nexus 870, Waltham, MA, USA) and XPS (Thermo Scientific K-Alpha, Waltham, MA, USA) were used to study the structure and composition of functional groups and chemicals. VSM (SQUID-VSM, San Diego, CA, USA), BET (Quadrasorb Si-3MP, Boynton Beach, FL, USA), and TGA (Tg-DTA 7300, Tokyo, Japan) were used to investigate the magnetic intensity, specific surface area, and thermal stability of hollow imprinted polymers before and after preparation, respectively. HPLC analysis was conducted on the Purkinje L600 HPLC system (Beijing, China) with a UV-Vis absorption detector.

### 2.2. Preparation of Core-Shell Hollow Magnetic Molecularly Imprinted Polymers

#### 2.2.1. Preparation of Core-Shell Magnetic Molecularly Imprinted Polymers

As shown in Figure 1, according to our previously published method, magnetic Fe_3_O_4_ nanoparticles and Fe_3_O_4_@mSiO_2_ particles were successively prepared and modified with vinyl [21,22] with MPS as follows. First, FeCl_3_·6H_2_O (1.35 g) and sodium acetate (3.60 g) were dispersed into ethylene glycol (20 mL) and ultrasonicated for 30 min. Similarly, the polyethylene glycol (1.00 g) was dispersed into ethylene glycol (20 mL) and ultrasonized for 30 min. The two mixed solutions were combined and ultrasonized again for 30 min. The resultant yellow solution was transferred to an autoclave lined with polytetrafluoroethylene, and after sealing, the solution was heated to 200 °C and reacted for 8 h. The final product was repeatedly washed with water and ethanol and dried in a vacuum-drying oven at 50 °C to obtain black Fe_3_O_4_ nanoparticles.

Then, the obtained Fe_3_O_4_ nanoparticles (100 mg) and CTAB (1.00 g) were added to 200 mL deionized water. CTAB was used to form a mesoporous structure in the SiO_2_ layer on the surface of ferric oxide. After mixing fully, 1.0 mmol/L NaOH solution (900 mL) was slowly added using a constant pressure funnel. After mixing again, the particles were transferred to a water bath at 60 °C for mechanical stirring and balancing for 30 min. Then, 5 mL TEOS/ethanol (1/4, *v*/*v*) was slowly added and stirred continuously at 60 °C for 24 h. Fe_3_O_4_@CTAB/SiO_2_ was collected by a magnet and then dispersed in acetone solution (180 mL) at 80 °C until the CTAB was removed, followed by vacuum drying at 50 °C to obtain Fe_3_O_4_@mSiO_2_ nanoparticles. The dried Fe_3_O_4_@mSiO_2_ (250 mg) and MPS (150 μL) were dispersed in 40 mL 10% acetic acid solution, stirred mechanically at 50 °C for 5 h, then collected by magnetic separation, washed repeatedly with deionized water, and dried under vacuum to obtain Fe_3_O_4_@mSiO_2_ modified by double bond. This modification step is shown in Figure 1 using a gradient color arrow.

After that, the template molecule GA (0.25 mmol) was dissolved in anhydrous acetonitrile. After ultrasonic dissolution, functional monomer 4-VP (1 mmol) was added under a nitrogen atmosphere, encapsulated, and stored at 4 °C for 12 h as the pre-assembly solution. Then, Fe_3_O_4_@mSiO_2_ (50 mg), AIBN (20 mg), and EGDMA (5 mmol) were dissolved in anhydrous acetonitrile. Under the condition of an ice-water bath, the pre-assembly solution was added under a nitrogen atmosphere for protection, and then transferred to a water bath at 60 °C with mechanical stirring for 24 h for the polymerization reaction. Finally, MMIP was obtained after reflux elution with methanol–acetic acid (9/1, *v*/*v*).

Molecular non-imprinted polymers (MNIP) were obtained by duplicating all steps without adding template.

#### 2.2.2. Preparation of Hollow Magnetic Molecularly Imprinted Polymers

To avoid excessive corrosion of the core Fe_3_O_4_, the conditions of preparing HMMIP were improved by optimizing the concentration of HF and the etching time, based on the report by Song et al. [23].

MMIP (10 mg) was dispersed in 65 μL HF solution (40 mmol/L) in deionized water, then oscillated in a shaker under 20 °C for 2 h. After solid–liquid separation with an external magnet, washing with water five times, and drying overnight under vacuum at 50 °C, HMMIP was obtained.

### 2.3. Characterization and HPLC Analysis

HPLC analysis was conducted on the Beijing Purkinje L600 HPLC system with a UV-Vis absorption detector (Purkinje, Beijing, China). The determination was performed on a Pgrandsil-STC-C_18_ column (4.6 mm × 150 mm, 5 μm) at a flow rate of 1.0 mL/min with an injection volume of 20 μL and a column temperature of 30 °C.

The ratio of the mobile phase and detection wavelength were set differently for different analytes, as shown in Appendix A.

### 2.4. Adsorption Performance

The process of the adsorption experiment is shown in Appendix A. A certain amount of polymer was dispersed in the solution, and after adsorption, it was separated by an external magnet. The supernatant was taken for HPIC analysis.

The adsorption capacity *Q*_t_ (mmol/g) at different contact time *t* (min) in the whole adsorption experiment was calculated by Equation (1):*Q*_t_ = (*c*_0_ − *c_t_*) *V*/*m*(1)
where *c*_0_ (mmol/L) and *c_t_* (mmol/L) are the initial concentration and the concentration at different moments of the target substance, respectively; *V* (L) is the volume of the target substance solution; and *m* (g) is the mass of HMMIP or MMIP.

#### 2.4.1. Kinetics Adsorption Experiment

For kinetic adsorption experiments, HMMIP/MMIP (30.0 mg) was mixed with 60 mL GA (2.0 mg/mL) in water. The mixtures were continuously shaken under 150 rpm at 318 K and the concentrations of GA at a certain interval (20, 30, 40, 50, 60 to 210 min) were analyzed by HPLC, and then the adsorption capacity *Q*_t_ (mmol/g) at different contact time *t* (min) was calculated by Equation (1).

#### 2.4.2. Thermodynamic Adsorption Experiment

For thermodynamic adsorption experiments, HMMIP/MMIP (10.0 mg) was suspended in GA aqueous solution (2.0 mL) with initial concentrations of 2.0 to 12.0 mg/mL. After shaking under 150 rpm at 298 K, 308 K, and 318 K for 3 h, the supernatant was detected by HPLC and the equilibrium adsorption capacity *Q*_e_ (mmol/g) of the GA was calculated similarly to *Q*_t_ based on Equation (1).

#### 2.4.3. Selective Adsorption Experiment

The selectivity of adsorption was performed using GA and its three structural analogues (salicylic acid, benzoic acid, and 2,4-dihydroxybenzoic acid) in standard solutions with initial concentrations of 12 mmol/L, 13 mmol/L, 14 mmol/L, and 16 mmol/L. The excessive concentration of structural analogues is more conducive to demonstrate the superior ability of the specific adsorption of HMMIP. The structures of four analogues are shown in Appendix A, in which the same color was used to mark the same parts between one another. HMMIP/MMIP/MNIP (10.0 mg) was suspended in standard aqueous solution (2.0 mL) with the same initial concentrations (12 mmol/L) of GA and its three structural analogues. After shaking under 150 rpm at 318 K for 3 h, the supernatants were detected by HPLC and the equilibrium adsorption capacity *Q*_e_ (mmol/g) of the four standard substances was calculated similarly to *Q*_t_ based on Equation (1).

The specific recognition of the HMMIP/MMIP was evaluated by imprinting factor (*α*) [20], which was calculated by Equation (2): *α* = *Q*_HMMIP or MMIP_/*Q*_MNIP_(2)
where *Q*_HMMIP or MMIP_ and *Q*_MNIP_ are the GA adsorption capacity (mmol/g) of the imprinted polymers and the corresponding non-imprinted polymers, respectively.

The selectivity of recognition was identified by the separation factor (*β*) [22], which was calculated by Equation (3):*β* = *α*_GA/_*α*_other_(3)
where *α*_GA_ is the imprinting factor for GA and *α*_other_ is the imprinting factor for structural analogues.

#### 2.4.4. Real Sample Analysis

One gram of green tea (Huangshan, China) was added to 80 mL distilled water for reflux extraction at 90 °C 3 times to obtain green tea solution.

HMMIP/MMIP (50 mg) was added to 2 mL of the above green tea solution, and shaken at 318 K for 150 min. After magnetic separation, the adsorbed HMMIP/MMIP was washed with 50 mL distilled water 3 times. Then, the adsorbed HMMIP/MMIP was eluted with methanol and acetic acid (9/1, *v*/*v*, 50 mL) to get the analyte. After the solvent was dried, the resultant analyte was dissolved in 1 mL distilled water for HPLC analysis. The HPLC analysis was performed using the same conditions as in Section 2.3.

## 3. Results and Discussion

### 3.1. Characterization of HMMIP and MMIP

#### 3.1.1. Analysis of FT-IR and XPS Results

In order to prove that the desired functional groups were grafted onto the corresponding materials and to analyze the etching of Si elements, the Fe_3_O_4_ nanoparticle, HMMIP, and MMIP were characterized by FT-IR. In addition, the HMMIP and MMIP were characterized by XPS spectroscopy. Meanwhile, the peak areas of Si element content were compared to each other.

As shown in Figure 2a, the FT-IR spectrum of MMIP showed that the peaks at 1077 cm^−1^ and 463 cm^−1^ were the asymmetrical stretching vibration peaks of Si-O, indicating that mSiO_2_ enveloped the surface of Fe_3_O_4_ successfully, and the absorption peak at 571 cm^−1^ was the vibration characteristic peak of Fe-O, which is consistent with that of the spectrum of Fe_3_O_4_ (Figure 2a). However, HMMIP did not have any stretching vibration peak at 1077 cm^−1^ or 463 cm^−1^, but the characteristic peak of Fe-O vibration at 571 cm^−1^ was retained, indicating that mSiO_2_ in HMMIP was effectively etched and the magnetic core was retained. In addition, for both HMMIP and MMIP, the stretching band at 1149 cm^−1^ was the characteristic peak of C–O–C, and 1637 cm^−1^ and 1728 cm^−1^ were the characteristic peaks of C=C and C=O, respectively. These were derived from AIBN, 4-VP, and EGDMA used in the polymeric imprinting layer, which indicates that the imprinted polymer covered the outermost surface very well. The similar peaks of the above functional groups have been reported in some references [24,25,26]. Thus, it could be preliminarily concluded that the mSiO_2_ layer in the middle of MMIP was effectively etched and the polymer layer was preserved.

From the XPS full survey scan (Figure 2b), Fe, O, N, C, and Si elements were detected in both HMMIP and MMIP, and similar positions of the binding energy of the same elements were reported in previous reports [27,28,29,30,31]. Compared with MMIP, the position of the binding energy peak of each element in HMMIP did not change, indicating that the etching of the mSiO_2_ intermediate layer did not affect the main structure or the properties of other layers of MIP. The Fe2p peak of HMMIP at 710.5 eV was derived from Fe^2+^ and Fe^3+^ in Fe_3_O_4_. However, in the full survey scan of MMIP, the Fe2p peak at 710.5 eV was smaller than that of HMMIP. The reason might be that Fe_3_O_4_ served as the core and mSiO_2_ as the intermediate layer of MMIP, and the X-ray emitted by the electron gun needed to penetrate more deeply.

In the full survey scan, the binding energy at 532.1 eV was attributed to the peak of O1s, which was derived from the oxygen elements in Fe_3_O_4_ and mSiO_2_ and Fe-O-Si. The N1s peak of 400 eV and C1s peak of 284.8 eV were obtained from AIBN, 4-VP, and EGDMA, which were used in the preparation of the imprinted polymers. Due to the very low content of N used in the polymerization reaction, the N1s peak was almost horizontal in the full survey scan. The peaks of Si2p at 102.9 eV were all attributed to SiO_2_. It can be seen from the full survey scan that the Si2p peak of HMMIP nearly disappeared after etching, indicating again that the intermediate layer mSiO_2_ in MMIP was successfully etched to form HMMIP.

If the O1s peak were fitted by peak separation to quantify the decrease of SiO_2_ content, the fitting results would be affected by C-O-C, C=O, and Fe-O, resulting in large errors. Therefore, the Si2p peak was selected for single-peak fitting of its spectrum (only Si-O is available). It can be seen from the fitting comparison of Si2p (Figure 2c) that after HF acid etching for 2 h, the Si content decreased by 81% according to the value of peak area, and decreased by the normalized method. Because HF acid can also react with Fe_3_O_4_ slowly, an optimized etching time of 2 h was enough to form a hollow structure with double imprinting surfaces. The remaining 19% of silicon is thought to be the result of uneven etching of a few polymers. Even if the etching time were prolonged, the adsorption capacity could not be significantly improved, and the Fe_3_O_4_ content would decrease, which would affect the magnetic response strength to delay magnetic separation.

In summary, combined with the spectra of FT-IR and XPS, it could be concluded that the core-shell MMIP was prepared smoothly, and mSiO_2_ was successfully etched to form HMMIP in a hollow structure with double imprinting surfaces.

#### 3.1.2. Analysis of TEM and Element Mapping

In order to further verify whether there was hollow structure in HMMIP, TEM detection was conducted for Fe_3_O_4_, Fe_3_O_4_@mSiO_2_, MMIP, and HMMIP; elemental mapping was carried out for MMIP and HMMIP; and the size of the hollow structure was measured using Image J software.

Figure 3A,B are TEM images of Fe_3_O_4_ and Fe_3_O_4_@mSiO_2_ at a 200 nm scale, respectively, with local enlarged images (50 nm scale) added. It was obvious that the size of the prepared Fe_3_O_4_ was relatively uniform (220 nm in diameter) in spherical morphology with excellent dispersion, and there was no adhesion to each other, which is consistent with the morphology in other reported research [32]. In Figure 3B, it can be clearly seen that the mSiO_2_ was successfully enveloped onto the nanoparticle of Fe_3_O_4_ to produce Fe_3_O_4_@mSiO_2_ particles with a uniform size of 290 nm in diameter. Some Fe_3_O_4_@mSiO_2_ particles looked like they adhered to each other, which could have been caused by insufficient ultrasonic processing during sample preparation.

Although the outermost imprinted polymer layer of MMIP was thin (Figure 3C), the light transmittance was too poor to observe the inner-layer structure. In order to determine the enveloping status of the polymer layer and the hollow structure in MMIP, the element superposition mapping of C (in yellow, Figure 3D) in the outermost surface of imprinted polymer, Si (in purple, Figure 3E) in the intermediate layer of mSiO_2_, and Fe (in green, Figure 3F) in the magnetic core of MMIP was performed.

First, it can be clearly seen that the Fe element was surrounded by a layer of Si (Figure 3G), since Fe_3_O_4_ was enveloped by the intermediate layer mSiO_2_. In the mapping of the Si element (Figure 3E), the middle part in light purple and the surrounding part in darker purple represent the thickness of the mSiO_2_ shell layer. Three parts of the circle were selected to measure the thickness of the mSiO_2_ shell layer with Image J software and the average value was obtained (T_1_ = 73.510 nm, T_2_ = 79.836 nm, T_3_ = 78.041 nm, T_average_ = 77.029 nm). This mSiO_2_ shell layer was thicker and denser than that we had previously studied [21], which may have been caused by the longer hydrolysis reaction time after the addition of TEOS in this study. Meanwhile, the imprinted polymer on MMIP’s outer layer was so thin that the interface between the imprinted polymer and the mSiO_2_ could not be clearly identify in the TEM images. 

It was obvious that HMMIP had a hollow structure if MMIP was etched with a higher HF concentration of 80 mmol/L, 120 mmol/L, 160 mmol/L, and 200 mmol/L, as shown in Figure 3H–K.

Unfortunately, Fe_3_O_4_ also could be etched partly due to higher HF concentration. In addition, the size of each HMMIP was not absolutely the same as before etching, which was caused by the different rates of HF entering the interior of the particle through mesoporous channels. Thus, Fe_3_O_4_ was etched in an irregular shape and its diameter was greatly reduced, which resulted in reducing its magnetic response intensity and prolonging its separation time from the solution after the target adsorbed. In addition, due to the rapid disappearance of mSiO_2_, there was obvious collapse and contraction of this MIP layer in this process, and the thickness of the MIP layer increased. Perhaps this is the reason why the MIP layer cannot be seen in Figure 3C, but can be clearly seen in Figure 3H–K. Its morphology was no longer a regular spheroid, which was also affected by intense erosion processes. Regarding the original intention of this article, the hollow structure itself would not definitely improve the adsorption capacity, but double MIP surfaces would play a great role in adsorption.

Therefore, 40 mmol/L HF was chosen as a mild etching concentration, the TEM images of HMMIP (Figure 4A,B) were very similar to those of Fe_3_O_4_, and the outermost imprinted polymer layer of HMMIP was thin, but its light transmittance was too poor to observe the interior structure. The element mapping of HMMIP (Figure 4C–E) shows that the core Fe_3_O_4_ (in red) was wrapped in an imprinted polymer layer (in green) without the Si element (in purple), compared with that in Figure 3G, which was caused by the etching of mSiO_2_.

The morphology of HMMIP was spherical with uniform size (about 300 nm in diameter), and the core Fe_3_O_4_ was not obviously etched (Figure 4E), which indicates that its magnetic response intensity would not be weakened under this etching condition (as proven in Section 3.1.4). However, the hollow structure could not be seen directly, as shown in Figure 3H–K, due to poor transmittance of the outmost MIP layer.

In order to calculate the gap between the core Fe_3_O_4_ and the MIP layer (the calculation principle is shown in Appendix A), Image J software was used to measure the thickness T_MIP_ of the MIP layer (in green in Figure 4E). Three sites from its element-mapping image (Figure 4G) were selected to calculate the average value of T_MIP_ (T_1_ = 6.273 nm, T_2_ = 6.074 nm, T_3_ = 5.959 nm, T_average_ = 6.012 nm), the diameter of HMMIP (D_HMMIP_) was measured as 289.777 nm, and the diameter of the Fe_3_O_4_ core was 270.894 nm. Thus, the width of the hollow layer in HMMIP could be calculated as 3.429 nm, which is much lower than the thickness of SiO_2_ and is believed to have been caused by the relative shrinkage of MIP.

Subsequent results of the research on adsorption performance also proved that the hollow structure actually existed. Compared with MMIP, the higher adsorption capacity of HMMIP attributed to both inner and outer adsorption surfaces demonstrates that the MIP layer remained intact, but only the mSiO_2_ layer was etched by the HF solution under the optimal conditions mentioned above.

#### 3.1.3. Determination of Specific Surface Area and Mesoporous Pore Size

The specific surface area, total pore volume, and average pore diameter are other intuitive physical data to characterize the adsorption performance of adsorption materials. Therefore, the BET specific surface area was measured for the prepared HMMIP, MMIP, and MNIP. The nitrogen adsorption–desorption curves (Appendix A) of the three MIPs were obtained by the BET method, and Appendix A shows the specific surface area, average pore size, and total pore volume measured by the Barrett–Joyner–Halenda method (BJH).

It should be noted that the pore structure of carbon materials is relatively complex, and it is easy to have flexible holes or ink bottle-shaped holes. After N_2_ adsorption, the pore diameter shrinks, which leads to the adsorption gas not being easy to desorb, and it is easy to cause the adsorption and desorption curve to not be closed (Appendix A).

It was obvious that the parameters of MMIP were not much improved compared with MNIP, which is also the weakness of traditional core-shell magnetic molecularly imprinted polymers. For example, the traditional core-shell magnetic molecularly imprinted polymers Fe_3_O_4_@MIPs prepared by Wang et al. [33] were found to have a specific surface area of only 8.31 m^2^/g and a total pore volume of 0.04 cm^3^/g by the BET method. Compared with MMIP, the average pore diameter of HMMIP in this paper was larger, the total pore volume increased by 2.89 times, and the specific surface area increased from 39.06 m^2^/g to 97.59 m^2^/g.

The HMMIP imprinted polymer layer was found to shrink to some extent after etching from MMIP in the above TEM analysis. It seems that the specific surface area should have decreased, but in fact the mesoporous channels expanded (as shown in Appendix A). This result indicates that the polymer layer had a certain toughness, which might have been caused by the addition of EGDMA as a cross-linking agent to improve its mechanical strength. In addition, the specific surface area was greatly increased because of the internal and external space.

The size of the pore diameter and the total pore volume were the key to forming the double adsorption surfaces. The average pore diameter of HMMIP was 6.74 nm, which enabled the outer adsorption surface to contact the target molecule after the HMMIP fully mixed with the solution. At the same time, the target molecule could also be adsorbed by the inner imprinted surface by crossing the mesoporous channel from the outer imprinted surface.

#### 3.1.4. Analysis of Magnetic Response Intensity of HMMIP

As for Appendix A, the shape of four curves of Fe_3_O_4_, Fe_3_O_4_@mSiO_2_, MMIP, and HMMIP was similar; there was no residual magnetism; the coercive force was zero; and all of them had superparamagnetic property. The saturation magnetization values of Fe_3_O_4_@mSiO_2_ (45.1 emu/g) and MMIP (48.1 emu/g) were lower and close to each other, indicating that the magnetic response of Fe_3_O_4_ was influenced by the covered non-magnetic mSiO_2_ shell layer. In addition, the imprinted polymer layer contained in MMIP was thin and had abundant mesoporous pores, so its influence on the saturation magnetization intensity could be ignored. The saturation magnetization of HMMIP (62.8 emu/g) was lower than that of the Fe_3_O_4_ nanoparticles due to the appearance of the nonmagnetic shell on the surface and the reaction between HF acid and Fe_3_O_4_ when etching, which had a certain influence on the magnetic response. However, the saturation magnetization of HMMIP was higher than that of Fe_3_O_4_@mSiO_2_ and MMIP due to the absence of mSiO_2_.

The imbedded image on bottom right of Appendix A shows the magnetically controlled separation of the HMMIP in the presence of an external magnetic field. HMMIP slowly settled down without an external magnet, but the HMMIP immediately (within 10 s) aggregated to the bottom of the bottle under an external magnet. Similar to MMIP, HMMIP could be separated quickly from the sample solution under an external magnetic field.

#### 3.1.5. TGA Analysis

The thermogravimetric analysis results of MMIP and HMMIP (Appendix A) implied that the mass loss of HMMIP was smaller than that of MMIP in the process of temperature rising. The reason might be that the mSiO_2_ layer was more sensitive to being heated than the molecularly imprinted layer. Thus, HMMIP was more thermally stable than MMIP.

In N_2_ atmosphere, when the temperature reached 200 °C, the mass loss of HMMIP and MMIP was only 1.02% and 2.52%, respectively, which was mainly due to the loss of water evaporation in the two samples. In the heating process at 400–500 °C, the mass of both decreased rapidly, which was caused by the decomposition of the imprinted polymer. The final weight loss of HMMIP was 12.63% when the temperature continued to rise to 900 °C. Therefore, the prepared HMMIP had higher thermal stability, which was not only superior to the MMIP prepared in this paper, but also superior to some previous reports on MIP [25,34,35,36,37].

According to the series of characterizations on HMMIP mentioned above, it can be concluded that mSiO_2_ was successfully etched, mesoporous channels were abundant, the specific surface area was larger, and inner and outer double adsorption surfaces were formed. Subsequently, GA was taken as the adsorbate to verify the actual adsorption capacity and application value of HMMIP.

### 3.2. Adsorption Performance of HMMIP

#### 3.2.1. Kinetics of Adsorption

The adsorption kinetics curves of HMMIP, MMIP, and MNIP at 318 K are shown in Appendix A. The adsorption capacity of three adsorbents for GA increased with the increment of adsorption time, increasing especially sharply starting at 50 min. The adsorption process of molecules permeated into the channel to identify the adsorption site was so slow that the adsorption process of MMIP reached equilibrium at 180 min, whereas HMMIP reached adsorption saturation at about 150 min. This could be attributed to the increase in the specific surface area for HMMIP when the intermediate mSiO_2_ layer was etched. The contact area with GA molecules on HMMIP was larger than that on MMIP, so the time to reach adsorption equilibrium was reduced. In contrast, the adsorption process of MNIP reached equilibrium at about 60 min with little adsorption capacity due to non-specific adsorption.

In order to investigate and compare the adsorption mechanisms of HMMIP, MMIP, and MNIP, the adsorption kinetic data at 318 K were used to fit the quasi-first-order kinetic Equation (4) and the quasi-second-order kinetic Equation (5).
ln(*Q*_e_ − *Q*_t_) = ln*Q*_e_ − k_1_*t*(4)
*t*/*Q*_t_ = 1/(k_2_*Q*_e_) *+ t*/*Q*_e_(5)
where *Q*_e_ (mmol/g) and *Q*_t_ (mmol/g) represent the adsorption amounts of GA at adsorption equilibrium and at time *t* (min), respectively, and k_1_ (min^−1^) and k_2_ (g mmol^−1^ min^−1^) are the adsorption rate constants.

The fitting points of the quasi-second-order kinetic fitting graphs of HMMIP (Appendix A), MMIP (Appendix A), and MNIP (Appendix A) fell on the fitting line, and the fitting correlation coefficients of all three MIPs were more than 0.999, whereas the fitting curves of the quasi-first-order kinetic equations showed poor correlation coefficients and scattered fitting points (Appendix A). The results indicate that the adsorption behavior of GA on HMMIP could be conformed to the quasi-second-order kinetic equation. The quasi-first-order and quasi-second-order kinetic adsorption data of HMMIP are analyzed in Appendix A, and the adsorption capacity calculated by the pseudo-second-order equation was closer to the measured adsorption capacity. Thus, the adsorption behavior of GA on HMMIP was in line with quasi-second-order equation.

Similarly, as for MMIP and MNIP, the adsorption capacities calculated by the pseudo-second-order equation was also closer to the measured adsorption capacities. Thus, the adsorption rate of GA on HMMIP, MMIP, and MNIP was positively correlated not only to the concentration of the adsorbate, but also to the amount of adsorbent. When the quantity of adsorbent was fixed, the growth of adsorption rate was limited by number of the adsorption sites on the adsorbent surface, so that there was a maximum adsorption rate.

#### 3.2.2. Thermodynamics of Adsorption

The adsorption capacity of the three molecularly imprinted polymers increased with the increment of temperature (Figure 5), The adsorption of GA on HMMIP and MMIP reached saturation adsorption when the initial concentration of GA increased to 10 g/L, whereas the adsorption of GA on MNIP reached saturation adsorption when the initial concentration of GA increased to 6 g/L. Perhaps the reason is that there are specific adsorption sites on HMMIP and MMIP, whereas there are only non-specific adsorption sites on MNIP. The saturated adsorption capacity of HMMIP was 2.815 mmol/g at 318 K, which was higher than those of MMIP and MNIP.

The adsorption of GA on traditional core-shell molecularly imprinted polymers, such as the GA-MMIP prepared by Zhan et al. [38], had a saturated adsorption capacity of 0.839 mmol/g. The saturated adsorption capacity for extracting other natural active ingredients by MMIP [39,40,41] was also lower than that of HMMIP, indicating that the HMMIP prepared in this study had higher adsorption efficiency compared with conventional core-shell molecularly imprinted polymers.

The Langmuir Equation (6) and Freundlich Equation (7) were also introduced to study whether the adsorption of GA on HMMIP, MMIP, and MNIP was a monolayer adsorption process or a multi-layer adsorption process, according to our previous paper [21]. The Langmuir equation was used to describe the monolayer adsorption process of adsorbents on the surface of the adsorbent, and the Freundlich equation was used to describe the multi-layer adsorption process.
(6)$$\frac{1}{Q_{e}}=\frac{1}{Q_{m}}+\frac{1}{Q_{m}K_{L}c_{e}}$$
ln*Q*_e_ = *m*ln*c*_e_ + ln*K*_f_(7)
where *Q*_e_ (mmol/g) is the adsorption capacity at adsorption equilibrium, *Q*_m_ (mmol/g) is the saturated adsorption capacity, and *c*_e_ (g/L) is the equilibrium concentration of GA. *K*_L_ is the parameter of the Langmuir isothermal equation, which indicates the ability of adsorption, and its value is related to the nature of the adsorbent, adsorbate, and temperature. *K*_f_ is the adsorption equilibrium constant of the Freundlich isothermal equation, which represents the adsorption capacity per unit concentration, whereas *m* is the Freundlich characteristic adsorption parameter.

The fitting correlation coefficients R_1_^2^ of the Langmuir equation with the temperature ranging from 298 K to 318 K for HMMIP were all greater than 0.999, higher than the fitting correlation coefficients R_2_^2^ of the Freundlich equation (Appendix A), indicating that the adsorption mode of GA on HMMIP highly matched the monomolecular layer adsorption. Moreover, the saturated adsorption capacity *Q*_m_(cal) calculated by the Langmuir equation was closer to the saturated adsorption capacity *Q*_e_(exp) measured in the experiment. Meanwhile, the fitting correlation coefficient R_1_^2^ of the Langmuir equation of MMIP and MNIP was also higher than the fitting correlation coefficient R_2_^2^ of the Freundlich equation, which indicates that the modification of MMIP (conversion to HMMIP) did not change its chemical environment, and the adsorption process of GA on HMMIP could be regarded as monolayer adsorption.

#### 3.2.3. Selectivity of Adsorption

Figure 6 shows the competitive adsorption of GA, salicylic acid, benzoic acid, and 2,4-dihydroxybenzoic acid on HMMIP, MMIP, and MNIP at 318 K. It can be seen that the adsorption capacity of GA on HMMIP was significantly higher than that of the other three substances. The imprinting factor *α*_HMMIP_ (3.180) was greater than that of *α*_MMIP_ (1.995), which also indicates that the specific adsorption capacity of GA on HMMIP was higher than that of the other three substances, and the specific recognition of GA on HMMIP was significantly stronger than that on MMIP. In addition, the values of the separation factor *β* for HMMIP calculated from Appendix A were almost all larger than for MMIP, indicating that the selectivity of recognition of HMMIP was higher than that of MMIP and could identify GA from structural analogues smoothly and efficiently. This is attributed to the fact that the imprinted cavity with GA as template was complementary to GA in shape, size, and functional groups.

As for the GA-MMIP synthesized by Zhang et al. [38], the imprinting factor *α* of GA-MMIP was 2.874 under the same calculation mode, which was lower than that of the HMMIP in this paper. Similarly, the separation factor *β* value of HMMIP was higher than that of the imprinted polymer HMICs prepared by Li et al. [42] for the adsorption of GA. Therefore, with high separation capability and selectivity, the HMMIP prepared in this paper could be used to extract GA from complex matrix.

#### 3.2.4. Specific Adsorption of Gallic Acid from Green Tea

After adsorption by HMMIP and MMIP, the GA peak height of adsorbed green tea solution decreased significantly in the HPLC data. Compared with the chromatogram of the green tea solution after adsorption on MMIP (Figure 7c), the peak of GA in the green tea solution after adsorption on HMMIP (Figure 7d) was especially low, indicating that the content of GA content was reduced and the specific adsorption capacity of HMMIP was superior to that of MMIP. According to the peak area normalization method, the adsorption efficiency of HMMIP and MMIP was 78.076% and 54.469%, respectively.

Furthermore, it can be seen that there was a series of miscellaneous peaks in the chromatogram of the green tea solution for 0–18 min (Figure 7a), and the height of these peaks did not decrease significantly in Figure 7c,d. In addition, the heights of other coexisting substances were similar after adsorption on MMIP and HMMIP (Appendix A), which indicates that the adsorption of GA on MMIP and HMMIP belonged to specific chemical adsorption. The adsorption capacity of GA on HMMIP increased more than on MMIP; nevertheless, physical adsorption slightly occurred for other substances. The chromatogram of the eluent from adsorbed HMMIP (Figure 7e) showed that except for the peak of GA, other miscellaneous peaks basically disappeared, which indicates that HMMIP had excellent separation and purification ability to specifically extract GA from complex matrix.

## 4. Conclusions

To sum up, this paper proposes a method to prepare hollow magnetic molecularly imprinted polymers (HMMIP). By using the method of surface imprinting polymerization, a common core-shell (MMIP) was synthesized with Fe_3_O_4_ as the core, mSiO_2_ as the intermediate layer, and imprinted polymer as the shell. Then, the intermediate mSiO_2_ layer was etched with HF acid to obtain HMMIP with a void in width of about 5 nm. The results of the investigation into the adsorption properties of HMMIP showed that the saturated adsorption capacity of GA on HMMIP was higher than that of GA on MMIP and MNIP. Under the same conditions, the saturated adsorption capacity of GA on HMMIP was significantly higher than that of other hydroxybenzoic acid derivatives on HMMIP, the recognition selectivity of HMMIP was higher than that of MMIP, and HMMIP could identify the GA from structural analogues smoothly and efficiently. Thus, the HMMIP prepared in this paper could be used to extract GA efficiently from complex matrix. Further optimization of the preparation conditions is needed to continuously improve the bonding fastness and adsorption performance of the HMMIP.

## Figures and Tables

**Figure 1 polymers-14-00175-f001:**
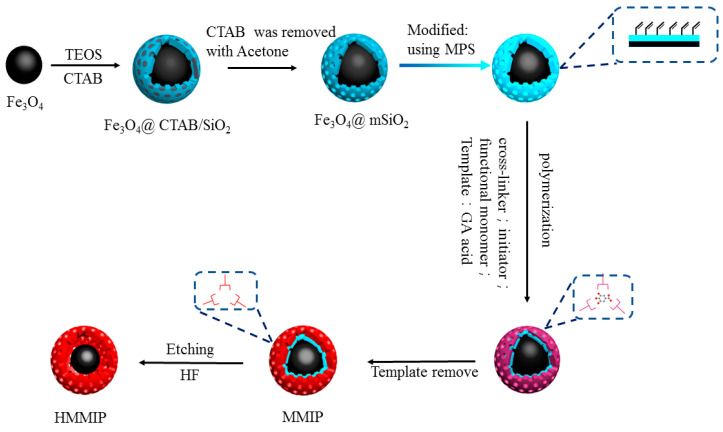
Synthesis of Fe_3_O_4_@Hollow MIP (HMMIP) by etching the SiO_2_ layer of Fe_3_O_4_@mSiO_2_@MIP (MMIP) using HF acid with GA as template.

**Figure 2 polymers-14-00175-f002:**
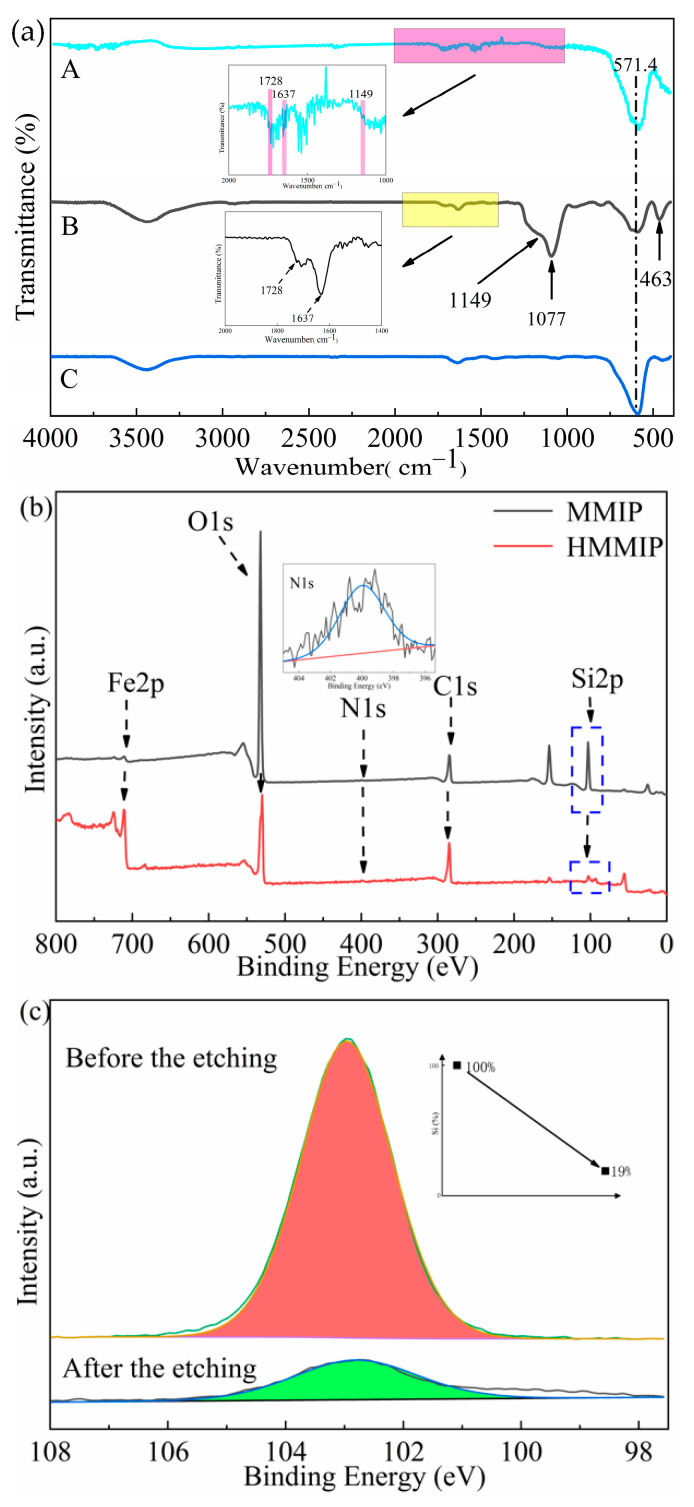
FT-IR spectra for HMMIP, MMIP, and Fe_3_O_4_ (**a**); XPS spectra for the full survey scan (**b**); and Si2p spectrum before and after HF etching (**c**).

**Figure 3 polymers-14-00175-f003:**
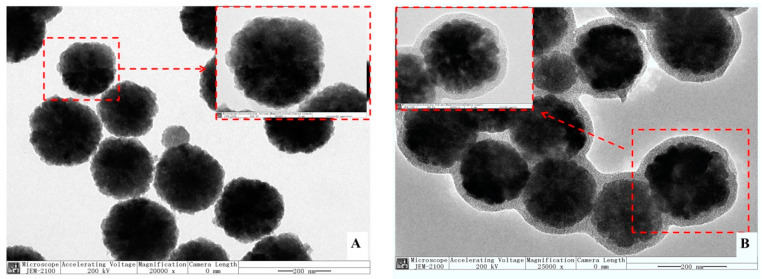
TEM images of Fe_3_O_4_ (**A**); Fe_3_O_4_@mSiO_2_ (**B**); MMIP (**C**); C, Si, Fe, and element superposition mapping of MMIP (**D**–**G**, respectively); HMMIP etched by 80 mmol/L (**H**); 120 mmol/L (**I**); 160 mmol/L (**J**); and 200 mmol/L HF (**K**); C, Fe, and element superposition mapping of HMMIP (**L**–**O**, respectively).

**Figure 4 polymers-14-00175-f004:**
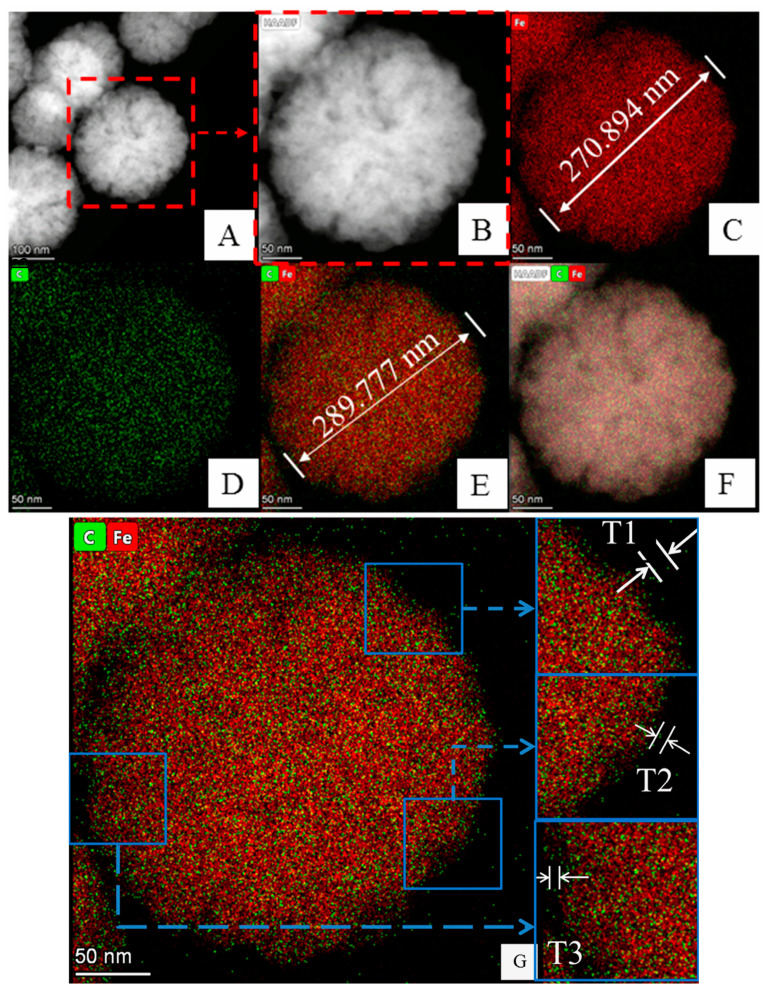
TEM images of HMMIP etched by 40 mmol/L HF (**A**,**B**) and element superposition mapping of HMMIP (**C**–**F**); HMMIP polymer shell thickness measurement (**G**).

**Figure 5 polymers-14-00175-f005:**
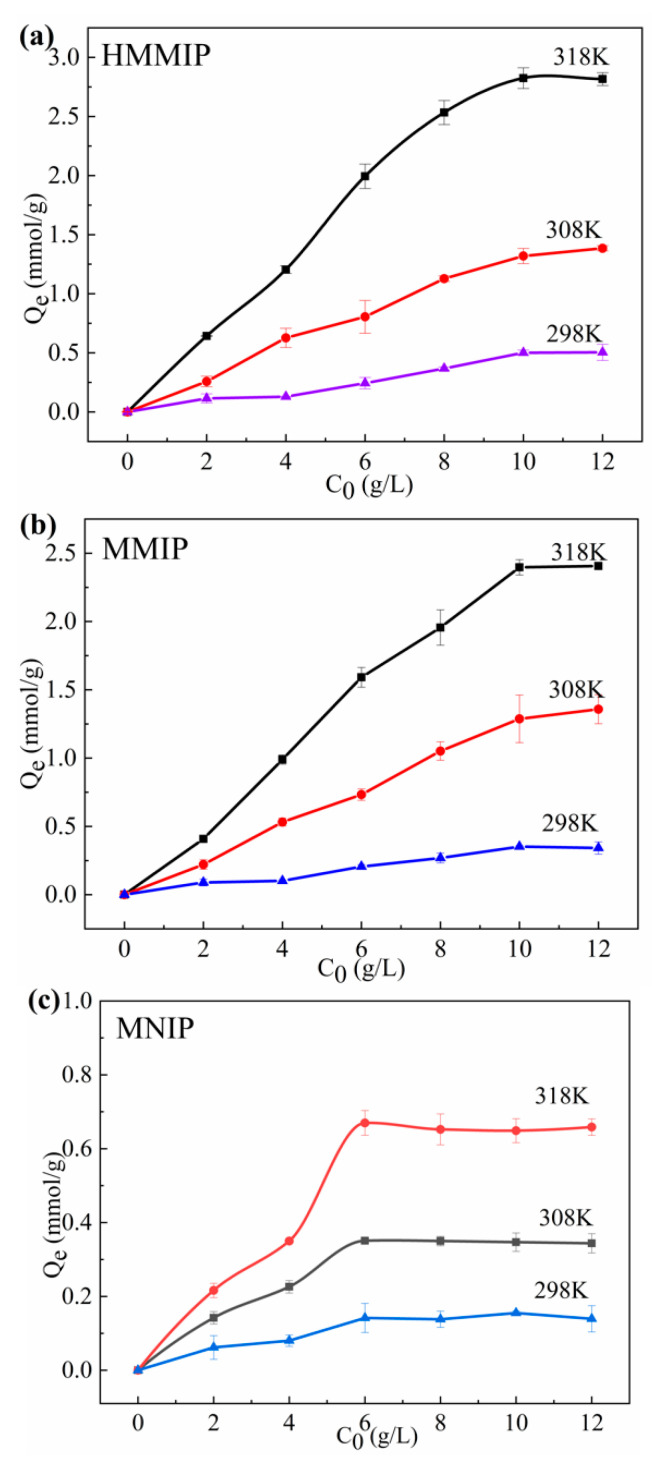
Adsorption thermodynamics curves of HMMIP (**a**), MMIP (**b**), and MNIP (**c**).

**Figure 6 polymers-14-00175-f006:**
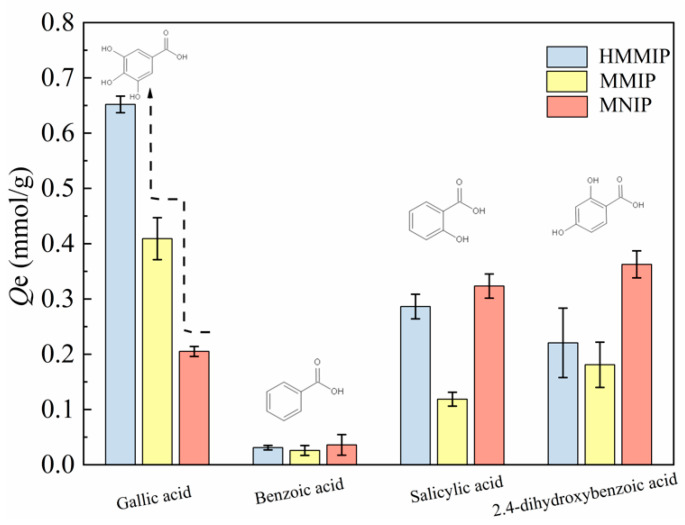
Competitive adsorption plots of HMMIP, MMIP, and MNIP at 318 K.

**Figure 7 polymers-14-00175-f007:**
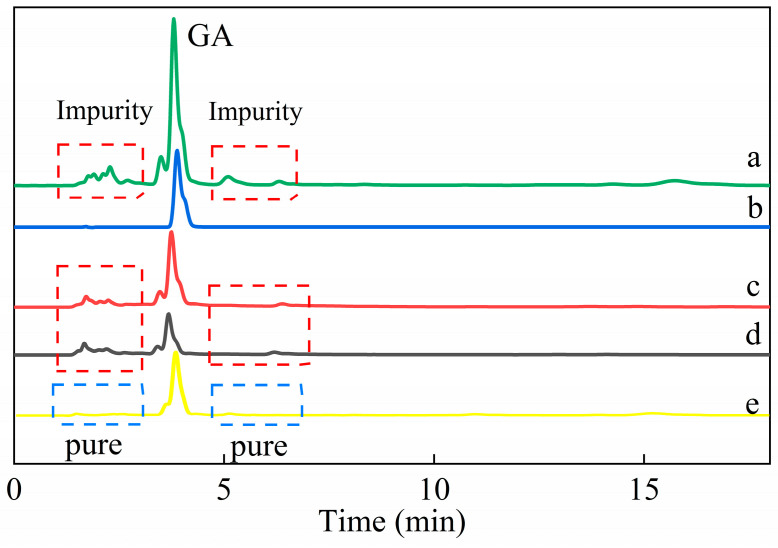
Chromatograms of the original green tea solution (**a**), standard gallic acid (**b**), the green tea solution after adsorption on MMIP (**c**) and HMMIP (**d**), and the eluent from the adsorbed HMMIP (**e**).

## Data Availability

Not applicable.

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
