# Peer review of "Selective Recognition of Gallic Acid Using Hollow Magnetic Molecularly Imprinted Polymers with Double Imprinting Surfaces"

_polymers, 2022, doi:10.3390/polym14010175_

Round 1
Reviewer 1 Report
Review: polymers-1524796.
Title: Selective recognition of gallic acid using hollow magnetic molecularly imprinted polymers with double imprinting surfaces.
In this research manuscript Authors presented the method for preparation of the hollow magnetic molecularly imprinted polymer (MIP) for selective adsorption of gallic acid. The characterization of two magnetic MIPs (viz. core-shell MIP and hollow MIP) and reference NIP materials was carried out in terms of adsorption characteristics, morphology and structure. The idea to obtain hollow structure of MIPs is not new but still has a potential for further studies to propose advanced materials. Nevertheless, the manuscript possesses some drawbacks that shall be addressed by Authors at this stage of evaluation:
- The novelty of the study should be clearly defined and indicated in the Introduction Section. The papers that refer to application of gallic acid analysis by MIPs and to synthesis of hollow MIP structures should be discussed to justify the novelty (see: J. Chromatogr. A 2021, 1637, 461829, Anal. Methods, 2018, 10, 3317, Food Chem. 2015, 179, 206, Food Chem. 2014, 146, 385, J. Agric. Food Chem. 2016, 64, 24, 5110, J. Pharm. Biomed. Anal. 2020, 180, 113036).
- The Introduction Section (lines 47-63) should be rewritten and shortened with brief reference to recent reviews of the field (see: Chem. Rev.2019, 119, 94, Materials 2021, 14, 1850, Eur. Polym. J. 2021, 145, 110231, Trends Anal. Chem. 2019, 114, 202, Talanta 2017, 167, 470, Front. Chem.2021, 9, 706311).
- The siloxane functionalization of magnetite prevents oxidation and aggregation of magnetic nanoparticles. Thus, the stability of magnetite after etching process should be investigated. Moreover, please provide data for non-imprinted hollow material since the studies are limited to comparison of core-shell MIP and NIP but not HM-NIP.
- Table S2. Nitrogen sorption analysis reveals nearly three-fold increase of the specific surface area of hollow MIP when compare to core-shell MIP (97.59 to 39.06 m2/g). One could expect that the increase should be doubled. Please provide data of the specific surface area for magnetite and magnetite functionalized only by silane layer. Since the hydrofluoric acid was used to etch silane layer, it could also promote the hydrolysis of the organic polymer chains – please comment. On the other hand, it could also indicate the adsorption on the magnetite surface. Thus, for that purpose, nitrogen sorption analysis should be also carried out for hollow NIP.
- Figure 6. Selectivity studies indicate that the hydroxy groups substituted to the aromatic ring play crucial role in the molecular recognition of the target analytes. In contrary, the carboxylic group role is rather limited. To confirm observation, the competitive studies shall include also molecules of phenol and catechol as reference compounds. Please note, that competitive adsorption studies should be carried out using equimolar amounts of target analytes (see lines 176 and 179).
- Experimental Section. The solid phase extraction (SPE) describes only loading and eluting steps. Why the washing step was omitted? Please comment, if the SPE process was optimized. Please provide SPE data for NIP since NIP could help to measure the non-specific adsorption. The recovery from hollow MIP was higher that from core-shell MMIP. Could Authors prove that the reason of phenomenon was related to double imprinting surface or to the non-specific adsorption on the magnetite surface?
- Editorial errors and punctuation mistakes in References should be avoided. Line 41 – should be type, line 42 – should be mechanism of MIP synthesis, line 46 – template is eluted only when non-covalent strategy is used, otherwise (covalent imprinting) the hydrolysis process should be applied to effectively remove the template, line 47 – I can not agree with Authors that the origins of the molecularly imprinting technology dated back to the 90’s. Please refer to the first paper of Prof. Wulff dated back to 1972. It should be also underlined that even earlier papers described the phenomenon of the molecular imprinting, line 57 – define phrase ‘conventional adsorbents’, line 59 – please correct the style. Caption of Figure 1 reveals that the schematic process is universal, viz. for MMIP and MNIP – thus, the indication of template over the arrow is misleading.
Based on above, I recommend major revision before final decision of the Editor.
Reviewer 2 Report
It is an interesting article. However you have to improve it
The English must be improved. I am giving you some examples:
Change
selectivity and saturated adsorption capacity (2.815 mmol/g at 318 K) of gallic acid on HMMIP were better than those on MMIP
with
the selectivity and the saturated adsorption capacity (2.815 mmol/g at 318 K) of gallic acid on HMMIP were better than those on MMIP
Change
Vlatakis et al [12] pioneered the use of "non-covalent imprinting" to prepare MIP, which were used to measure drug content in human serum
with
Vlatakis et al [12] pioneered the use of "non-covalent imprinting" to prepare MIPs, which were used to measure drug content in human serum
Change
The structure of HMMIP was characterized and the adsorption performance of HMMIP for GA was investigated through kinetic, thermodynamic, and specific adsorption experimental
with
The structure of HMMIP was characterized and the adsorption performance of HMMIP for GA was investigated through kinetic, thermodynamic, and specific adsorption assessments.
Change:
As showing in Figure 1
with
As shown in Figure 1
Change:
….and then transferred water bath at 60℃ for ….
with
…and then transferred to a water bath at 60℃ for …
I stop here! Please read again all the article and make corrections! Or give it to someone with very good English knowledge!
I do not understand the phrase bellow:
Elution of HMMIP was performed with methanol and acetic acid (9/1, V/V,50 mL). After the solvent was dried and eluted, the resultant sample was dissolved in 1 mL distilled water for HPLC analysis.
Please explain in experimental part how did you prepare the samples for XPS and TEM? Are there sections in the particles? Because XPS and TEM measure only on the surface!!
I do not see the green in figure 4 -E.
Please check if it is true : Three sites from its element mapping image (Figure 3-G) were selected to calculate the average value is not correct. You mean perhaps: Figure 4 _G. Please read again and make corrections in this part!
I do not understand this phrase:
Although the HMMIP imprinted polymer layer was found to shrink to some extent after etching from MMIP in above TEM analysis.
Round 2
Reviewer 1 Report
Review: polymers-1524796_R2.
Title: Selective recognition of gallic acid using hollow magnetic molecularly imprinted polymers with double imprinting surfaces.
In this revised manuscript, Authors have made corrections according to referee comments. In my opinion, the manuscript in current form could be considered for acceptance.
Reviewer 2 Report
Now the article is OK!